# Relative Biological Effectiveness of Carbon Ions for Head-and-Neck Squamous Cell Carcinomas According to Human Papillomavirus Status

**DOI:** 10.3390/jpm10030071

**Published:** 2020-07-25

**Authors:** Naoto Osu, Daijiro Kobayashi, Katsuyuki Shirai, Atsushi Musha, Hiro Sato, Yuka Hirota, Atsushi Shibata, Takahiro Oike, Tatsuya Ohno

**Affiliations:** 1Department of Radiation Oncology, Gunma University Graduate School of Medicine, 3-39-22, Showa-machi, Maebashi 371-8511, Japan; m12201018@gunma-u.ac.jp (N.O.); yukahirota@gunma-u.ac.jp (Y.H.); tohno@gunma-u.ac.jp (T.O.); 2Department of Radiation Oncology, Gunma Prefectural Cancer Center, 617-1, Takahayashi-nishicho, Ota 373-8550, Japan; m07201029@gmail.com; 3Department of Radiology, Jichi Medical University, 3311-1, Yakushiji, Shimotsuke, Tochigi 329-0498, Japan; kshirai@jichi.ac.jp; 4Gunma University Heavy Ion Medical Center, 3-39-22, Showa-machi, Maebashi 371-8511, Japan; musha@gunma-u.ac.jp (A.M.); hiro.sato@gunma-u.ac.jp (H.S.); 5Signal Transduction Program, Gunma University Initiative for Advanced Research (GIAR), 3-39-22, Showa-machi, Maebashi 371-8511, Japan; shibata.at@gunma-u.ac.jp

**Keywords:** carbon-ion radiotherapy, head-and-neck tumors, squamous cell carcinoma, radiosensitivity, relative biological effectiveness

## Abstract

Carbon-ion radiotherapy (CIRT) has strong antitumor effects and excellent dose conformity. In head-and-neck squamous cell carcinoma (HNSCC), human papillomavirus (HPV) status is a prognostic factor for photon radiotherapy outcomes. However, the effect of HPV status on the sensitivity of HNSCCs to carbon ions remains unclear. Here, we showed that the relative biological effectiveness (RBE) of carbon ions over X-rays was higher in HPV-negative cells than in HSGc-C5 cells, which are used for CIRT dose establishment, whereas the RBE in HPV-positive cells was modest. These data indicate that CIRT is more advantageous in HPV-negative than in HPV-positive HNSCCs.

## 1. Introduction

Carbon-ion radiotherapy (CIRT) is a promising modality with a strong antitumor effect even in tumors resistant to conventional photon radiotherapy [1]. CIRT is characterized by a highly conformal dose distribution to targets; this property enables sufficient dose delivery to tumors in the head-and-neck region while sparing the surrounding critical organs at risk (e.g., the salivary glands and the eyes) [2]. In Japan, which has the greatest number of CIRT facilities in the world, non-squamous cell carcinomas have been the major target for CIRT in the head and neck regions. However, radiation oncologists often encounter head-and-neck squamous cell carcinomas (HNSCCs) that show resistance to photon radiotherapy, underscoring the need to establish a strategy for stratifying patients and for switching photon-resistant HNSCC patients to CIRT.

Precision medicine enables the design of personalized treatment strategies according to the biological information about individual tumors [3]. Certain genetic profiles predict radiotherapy outcomes, suggesting that precision medicine can be applied to radiotherapy. For example, we and others showed that the *EGFR* mutational status predicts the sensitivity of non-small cell lung cancers to photon radiotherapy [4,5]. Torres-Roca et al. developed an algorithm based on the mRNA expression levels of ten genes to predict radiotherapy outcomes in various cancers [6]. Human papillomavirus (HPV) infection is a favorable prognostic factor in HNSCC [7], and HPV-negative HNSCCs are more resistant to photon irradiation than HPV-positive HNSCCs [8]. The molecular mechanisms explaining the difference in photon sensitivity between HPV-positive and negative HNSCCs have not been elucidated fully. However, studies suggest p53-associated apoptosis and DNA repair as candidate mechanisms [8]. For the former, it is considered that the function of p53 in inducing apoptosis is retained in HPV-positive head and neck cancer cells, although downregulated by the HPV E6-oncoprotein, whereas HPV-negative head and neck cancer cells harbor genetic alterations in *p53*, contributing to the anti-apoptotic phenotype. For the latter, data suggest that in HPV-positive cancer cells, p16 overexpression impairs the recruitment of RAD51 to DNA damage sites, contributing to decreased homologous recombination activity. HPV negativity thus has potential as a biomarker of photon resistance that can be used to stratify patients with HNSCC and to determine indications for CIRT. However, there is limited information on the sensitivity of HNSCCs to carbon ions. For example, Particle Irradiation Data Ensemble (PIDE, ver. 3.1), a large database of cell lines tested in vitro for sensitivity to particle radiation, contains only three HNSCC cell lines [9]. In addition, the effect of HPV status on the relative biological effectiveness (RBE) of carbon ions remains unclear. To address these issues, we examined the carbon-ion sensitivity of HNSCCs according to HPV status, as well as the RBE of CIRT in this patient population.

## 2. Materials and Methods

### 2.1. Cell Line and Cell Culture

Four HPV-positive and six HPV-negative HNSCC cell lines were used in this study (Table 1) [10,11,12,13,14]. UD-SCC-2 was obtained from Dr. Silke Schwarz of Ulm University (Ulm, Germany). UM-SCC-1, UM-SCC-47, and UM-SCC-104 were obtained from Merck (Kenilworth, NJ, USA). UPCI: SCC154 was obtained from Dr. Susanne Gollin of University of Pittsburgh (Pittsburgh, PA, USA). A-253, Detroit 562, Fadu, SCC-9, and SCC-25 were obtained from ATCC (Manassas, VA, USA). Based on the historical context of the development of CIRT, the HSGc-C5 cell line, obtained from JCRB Cell Bank (National Institutes of Biomedical Innovation, Health and Nutrition, Ibaragi, Japan), was used as the reference for the carbon-ion RBE [15]. The cells were cultured in RPMI-1640 (Sigma-Aldrich, St. Louis, MO, USA) supplemented with 10% fetal bovine serum (Life Technologies, Carlsbad, CA, USA).

### 2.2. Clonogenic Assays

Clonogenic assays were performed as described previously [16]. Briefly, cells thawed from frozen stocks were cultured over more than two passages and used for experiments after confirming that they were in the logarithmic growth phase. The cells were treated with trypsin (Sigma-Aldrich) for 5 min at 37 °C for detachment, and single-cell suspensions in culture media were prepared in 50 mL conical tubes. The cell numbers were determined using the improved Neubauer hemocytometer with an inverted microscope. Based on the cell count results, the seeding of the cells was performed using final cell suspensions prepared after two one-to-ten serial dilutions (×100 dilution in total). The cells seeded in 6-well plates were incubated for a minimum period to enable cell attachment (6–12 h, according to cell line characteristics) and received X-ray or carbon ion irradiation. After incubation for an additional 10–14 days, the cells were fixed with methanol and then stained with crystal violet. Clumps of equal to or more than 50 cells were recognized as colonies, and the number of colonies per well was determined using an inverted microscope. The surviving fraction for a given dose was calculated by dividing the number of colonies for the dose by the number of seeded cells for the dose, which was further divided by the plating efficiency, calculated based on unirradiated controls. The surviving fractions were fitted to the linear quadratic model [17], and the D_10_ (i.e., the dose that reduces cell survival to 10%) was calculated. The RBE of carbon ions for a given cell line was calculated by dividing the D_10_ value for X-rays by that for carbon ions [2,15]. At least four samples were used for each experiment. The experiments were repeated at least twice.

### 2.3. Irradiation

X-ray irradiation was performed using a Faxitron RX-650 (100 kVp, 1.14 Gy/min; Faxitron Bioptics, Tucson, AZ, USA) [4]. Carbon-ion irradiation was performed at the Gunma University Heavy Ion Medical Center. The specific parameters were as follows: 290 MeV/nucleon; an average linear energy transfer at the center of a 6 cm spread-out Bragg peak (SOBP) of approximately 50 keV/µm [4].

### 2.4. Statistics

Differences between two non-paired groups were examined using the non-parametric two-sided Mann–Whitney U-test. Differences between two paired groups were examined using the non-parametric two-sided Wilcoxon paired signed-rank test. The level of significance for the differences was set at *p* of <0.05. All statistical analyses were done by using Prism8 (GraphPad Software, San Diego, CA, USA).

## 3. Results

The sensitivity of the four HPV-positive and six HPV-negative HNSCC cell lines to X-rays was tested using clonogenic assays (Table 1). The results showed that resistance to X-rays was higher in HPV-negative than in HPV-positive cell lines (Figure 1a). D_10_ values were significantly higher for HPV-negative cell lines than for HPV-positive cell lines (8.2 ± 2.2 Gy vs. 4.3 ± 1.2 Gy, *p* = 0.038) (Figure 2a). These data are consistent with the findings of previous studies showing that HPV-negative head-and-neck tumors are resistant to X-rays compared with their HPV-positive counterparts and confirm the robustness of the present experimental system for the assessment of radiosensitivity [8].

Examination of the clonogenic survival of the ten cell lines treated with carbon-ion irradiation showed that carbon ions had a greater cell killing effect than X-rays in all cell lines examined (Figure 1a,b). The D_10_ values were significantly higher for HPV-negative cell lines than for HPV-positive cell lines (3.3 ± 0.7 Gy vs. 2.3 ± 0.3 Gy, *p* = 0.033) (Figure 2b). However, the D_10_ range was narrower for carbon ions than for X-rays, indicating the high cell killing effect of carbon ions on HNSCC cells regardless of HPV status (Figure 2a,b). The differences in α/β values between X-rays and carbon ions were not statistically significant (*p* = 0.46) (Appendix A).

The RBE of carbon ions was calculated in HPV-positive and HPV-negative cell lines (Table 2). The HSGc-C5 cell line was used as the reference for RBE because the clinical dose unit for CIRT, i.e., Gy (RBE), is determined using RBE data obtained with this cell line [15,18]. Under the experimental conditions used, the RBE in HSGc-C5 cells was 2.0 (Appendix A). The RBE was higher in all the six HPV-negative cell lines (i.e., 2.4 ± 0.2) than in the HSGc-C5 cell line (Figure 2c). By contrast, the RBE was lower in three of the four HPV-positive cell lines (i.e., 2.1 ± 0.8) than in HSGc-C5 cells (Figure 2c). These data indicate that CIRT is more effective in HPV-negative HNSCCs than in HPV-positive HNSCCs.

## 4. Discussion

We showed that the RBE of carbon ions over X-rays was higher in HPV-negative cells than in the HSGc-C5 cell line, which was used as the reference for CIRT dosing, whereas the RBE of carbon ions was lower in HPV-positive cells. This is the first report analyzing the RBE of carbon ions in relation to HPV status in HNSCC. To the best of our knowledge, this is the largest dataset of cells used for the in vitro testing of the sensitivity of HNSCC to carbon ions.

Full clinical studies for CIRT began in 1994 at the Heavy Ion Medical Accelerator in Chiba (HIMAC) of the National Institute of Radiological Sciences (NIRS) in Japan [1]. The Gesellschaft für Schwerionenforschung (GSI) center in Germany started treating patients with carbon ions in 1997. Two CIRT centers in Japan—i.e., the Hyogo Ion Beam Medical Center (HIBMC) and Gunma University Heavy Ion Medical Center (GHMC)—joined in 2002 and 2010, respectively, and the CIRT facilities are now spreading mainly in Asia and Europe. Nevertheless, compared with proton radiotherapy, another particle therapy modality, it can still be said that CIRT is an extremely limited medical resource because only 13 facilities in the world are in operation as of 2020. Annually, more than 14 million patients are newly diagnosed with cancer worldwide. By contrast, CIRT has the capacity of only a few thousand per year, even if all the facilities in the world were combined. This means that less than one percent of the patients newly diagnosed as having cancer can have the opportunity to be treated with CIRT. The biggest barriers to the development of a new CIRT facility are the high capital costs and the high operational costs: it is estimated that it costs approximately 138.6 million EUR [1]. Without a technological breakthrough to lower the costs’ magnitude, the critical shortage of CIRT facilities will remain for the coming decades. From this perspective, the optimization of patient stratification is of high importance to maximize the use efficacy of CIRT. In this context, the tumors located in the head-and-neck regions are ideal targets for CIRT, which possesses high organ-sparing ability, because the head-and-neck regions are one of the anatomical sites where functionally important organs are concentrated, forming a complex structure with each other. In addition to the advantages of carbon ions over photons in terms of dose conformity, high-linear-energy-transfer carbon ions show strong cell-killing effects even for photon-resistant cancer cells. Taking these together, in the present study, we sought to elucidate the RBE of carbon ions for a subset of photon-resistant head-and-neck tumors, to be considered as a candidate for CIRT.

When CIRT was launched at NIRS, the clinical SOBP of HIMAC beams was designed to achieve uniform cell killing across the SOBP width [15]. For this purpose, the a and b parameters in the linear quadratic survival curves corresponding to varying linear energy transfer were obtained by clonogenic assays. The cell line used as the reference in the series of clonogenic assay experiments should represent the responses of tumors to carbon ions. Therefore, HSG cells, which show intermediate radiosensitivity among various cell lines, were chosen. From this standpoint, although the current CIRT employs the dose unit of Gy(RBE), the antitumor effect of the CIRT compared with photon radiotherapy should be different according to the actual RBEs that the individual tumors show. Thus, it is critically important to elucidate the RBE of a subset of tumors that share specific biological features in order to achieve the above-mentioned optimization of patient stratification in CIRT. To this end, in the present study, we sought to elucidate the RBE of HNSCCs according to HPV status by using HSG cells as the reference.

Kagawa et al. reported that the RBE for carbon ions for HSGc-C5 cells was 1.8 at the center of a 6 cm SOBP of 320 MeV/n beams using 4 MV X-rays as the control [18]. Yoshida et al. reported that the RBE for HSGc-C5 cells was 2.0 at the center of an 8 cm SOBP of 350 MeV/n beams using 200 kVp X-rays as the control [19]. The RBE values reported in these studies are broadly consistent with that obtained in the present study (i.e., 2.0 at the center of a 6 cm SOBP of 290 MeV/n beams using 100 kVp as the control), supporting the robustness of our experimental systems for RBE measurements.

Mizoe et al. reported the outcomes of a phase II clinical trial of CIRT in 236 head-and-neck tumors [20]. In that study, 64.0 Gy(RBE) given in 16 fractions over 4 weeks caused modest normal tissue toxicities; i.e., grade 3 skin reactions in 6% and grade 3 mucosal reactions in 10% of the patients in the acute phase, and grade 2 skin reactions in 3% and grade 2 mucosal reactions in 2% of the patients in the late phase. In this study, the RBE values in HPV-negative cells were approximately 10–30% higher than those in HSGc-C5 cells. The clinical Gy(RBE) dose unit is calculated according to the clonogenic survival of HSGc-C5 cells. Therefore, theoretically, it is estimated that CIRT at 64.0 Gy(RBE) exerts antitumor effects equivalent to approximately 70.4–83.2 Gy(RBE) in HPV-negative HNSCCs with moderate tissue toxicities. These data indicate that CIRT may be beneficial in this type of tumor. Clinical trials of CIRT for the treatment of HPV-negative HNSCCs are necessary to validate the results of the present study.

UM-SCC-104 showed an RBE greater than 3, which was exceptionally high for an HPV-positive cell line. In fact, the D_10_ for X-rays for this cell line was intermediate (i.e., 6.5 Gy), ranking it fifth among the ten cell lines examined in this study. By contrast, the D_10_ for carbon ions for this cell line was the lowest among the ten cell lines (i.e., 1.9 Gy). This indicates a specific sensitivity of UM-SCC-104 to carbon ions. Wang et al. reported that in the repair of DNA double-strand breaks (DSBs) induced by heavy ions, only Ku-dependent non-homologous end joining (NHEJ), but not other DSB repair pathways, is impaired [21]. Tang et al. reported that upon the establishment of the UM-SCC-104, this cell line shows strong and diffuse expression of EGFR [22], a protein known to be involved in NHEJ by interacting with the catalytic subunit of DNA-dependent protein kinase [23]. These data together may suggest that UM-SCC-104 relies heavily on NHEJ in the repair of DSBs.

There was obvious difference in photon sensitivity between the HPV-positive and HPV-negative HNSCC lines examined in this study, whereas the difference in carbon ion sensitivity between the two groups was mediocre. It is worth noting that the HPV-positive lines harbored wild-type *p53*, whereas the HPV-negative lines were *p53*-mutant (Table 3) [24,25,26]. We have previously shown that photon sensitivity is affected by *p53* status; functional deficiency in p53 contributes to photon resistance through the abrogation of radiation-induced apoptosis. By contrast, carbon ions exert cell killing regardless of *p53* status through the efficient induction of mitotic catastrophe [27]. Thus, it can be hypothesized that CIRT shows promising anti-tumor effects for anti-apoptotic p53-mutated HPV-negative HNSCCs, which should be tested in future research.

One limitation of this study is that we did not analyze the molecular mechanisms underlying the higher RBE for HPV-negative HNSCCs, and this issue warrants future research.

## 5. Conclusions

We determined the RBE of carbon ions in HPV-positive and HPV-negative HNSCCs. The RBE values were higher in HPV-negative cells than in the reference HSGc-C5 cells, suggesting that CIRT is beneficial for the treatment of HPV-negative HNSCC and that clinical validation is warranted.

## Figures and Tables

**Figure 1 jpm-10-00071-f001:**
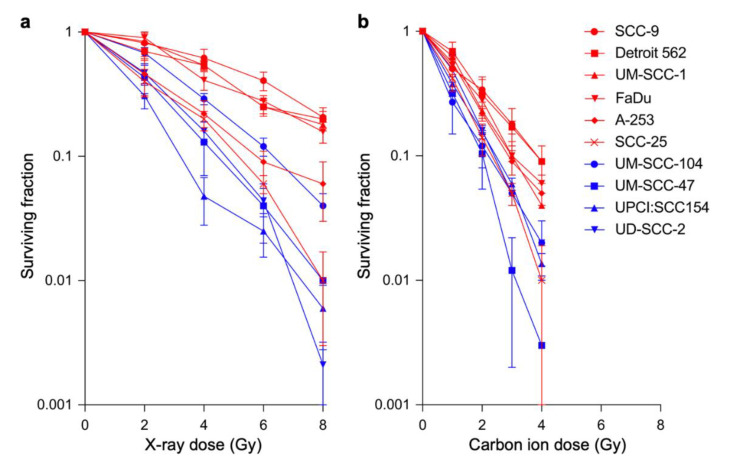
Clonogenic survival of head-and-neck squamous cell carcinoma cell lines treated with X-rays (**a**) or carbon ions (**b**). Human papillomavirus-positive or -negative cell lines are indicated in blue and red, respectively.

**Figure 2 jpm-10-00071-f002:**
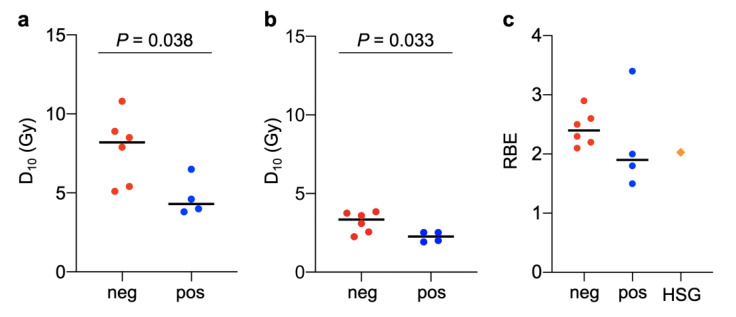
D_10_ for X-rays (**a**), D_10_ for carbon ions (**b**), and relative biological effectiveness (RBE) of carbon ions (**c**) for human papillomavirus-positive (pos) or -negative (neg) head-and-neck squamous cell carcinoma cell lines. RBE data for the HSGc-C5 cell line (HSG) were used as the reference. Black lines show median values. *p*-values as assessed by the Mann–Whitney U-test are shown.

**Table 1 jpm-10-00071-t001:** Cell lines used in this study.

Cell Line	HPV Status	Origin	Reference
UD-SCC-2	Positive	Pharynx	[10]
UM-SCC-47	Positive	Tongue	[11]
UM-SCC-104	Positive	Oral cavity	[10]
UPCI:SCC154	Positive	Tongue	[10]
A-253	Negative	Salivary gland	[12]
Detroit 562	Negative	Pharynx	[13]
FaDu	Negative	Pharynx	[13]
SCC-9	Negative	Tongue	[14]
SCC-25	Negative	Tongue	[13]
UM-SCC-1	Negative	Oral cavity	[11]

HPV, human papillomavirus.

**Table 2 jpm-10-00071-t002:** D_10_ and RBE in HPV-positive and negative cell lines.

Cell Line	HPV Status	D_10_-X (Gy)	D_10_-C (Gy)	RBE
UD-SCC-2	Positive	4.58	2.52	1.81
UM-SCC-47	Positive	3.98	2.01	1.98
UM-SCC-104	Positive	6.52	1.92	3.38
UPCI:SCC154	Positive	3.80	2.52	1.50
A-253	Negative	5.40	2.56	2.10
Detroit 562	Negative	8.48	3.84	2.20
FaDu	Negative	8.85	3.60	2.45
SCC-9	Negative	10.83	3.75	2.88
SCC-25	Negative	5.06	2.25	2.25
UM-SCC-1	Negative	7.90	3.09	2.56

X, X-rays; C, carbon ions.

**Table 3 jpm-10-00071-t003:** *TP53* mutation status for the cell lines used in this study.

Cell Line	*TP53*	Reference
UD-SCC-2	wild-type	[24]
UM-SCC-47	wild-type	[24]
UM-SCC-104	wild-type	[22]
UPCI:SCC154	wild-type	[25]
A-253	mutant (deletion)	[12]
Detroit 562	mutant (codon 248)	[12]
FaDu	mutant (codon 258)	[12]
SCC-9	mutant (deletion)	[12]
SCC-25	mutant (deletion)	[24]
UM-SCC-1	mutant (splice site)	[26]

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
