# Peer review of "Relative Biological Effectiveness of Carbon Ions for Head-and-Neck Squamous Cell Carcinomas According to Human Papillomavirus Status"

_jpm, 2020, doi:10.3390/jpm10030071_

Round 1
Reviewer 1 Report
The authors established cell survival curves in response to carbon ions and X-Rays for 10 HNSCC cancer cell lines of which 6 are HPV negative and 4 HPV positive. They also determined the RBE at 10% survival for each cell line. These data are original and will have to be confirmed in a clinical study. In my opinion, this article could be accepted with minor revisions:
- Introduction: The molecular mechanism explaining the difference in radiosensitivity of HPV positive and negative cells in response to photon irradiation should be recalled.
- Material and methods: According to the experimental protocol, it seems that the authors do not consider plating efficiency for the surviving fraction calculation as conventionally recommended. Could the authors comment on that?
- Results: The authors must draw up a table with the D10 and EBR values ​​for each cell line and discuss the results of the positive HPV cell line with an EBR greater than 3.
- Discussion: the authors should put forward hypotheses concerning the difference in behaviour of positive and negative HPV cell lines in response to photon and carbon ion irradiation. Do the authors have the p53 status of each cell line? These data would be important for interpreting the results.
Author Response
Reviewer #1
The authors established cell survival curves in response to carbon ions and X-Rays for 10 HNSCC cancer cell lines of which 6 are HPV negative and 4 HPV positive. They also determined the RBE at 10% survival for each cell line. These data are original and will have to be confirmed in a clinical study. In my opinion, this article could be accepted with minor revisions.
Response:
We sincerely thank the reviewer for evaluating our manuscript and for the encouraging comments. We made a thorough revision on our manuscript in accordance with the suggestions as follows.
- Introduction: The molecular mechanism explaining the difference in radiosensitivity of HPV positive and negative cells in response to photon irradiation should be recalled.
Response:
We thank the reviewer for the insightful comment. The molecular mechanisms explaining the difference in photon sensitivity between HPV positive and negative HNSCCs have not been elucidated fully. However, studies suggest p53-associated apoptosis and DNA repair as candidate mechanisms (Mirghani et al., Cancer Treat Rev 2015;41:844−852). For the former, it is considered that the function of p53 to induce apoptosis is retained in HPV-positive head and neck cancer cells, although downregulated by the HPV E6-oncoprotein, whereas HPV-negative head and neck cancer cells harbor genetic alteration in p53, contributing to anti-apoptotic phenotype. For the latter, data suggest that in HPV-positive cancer cells, p16 overexpression impairs the recruitment of RAD51 to DNA damage sites, contributing to decreased homologous recombination activity. This was added in the lines 46−54.
- Material and methods: According to the experimental protocol, it seems that the authors do not consider plating efficiency for the surviving fraction calculation as conventionally recommended. Could the authors comment on that?
Response:
We did consider plating efficiency to calculate surviving fraction, however, the explanation in the methods section was inappropriate. We apologize. To clarify the usage of plating efficiency in the calculation of surviving fraction, the texts were changed as follows: "The surviving fraction for a given dose point was calculated by dividing the number of colonies for the dose point by the number of seeded cells for the dose point, which was further divided by plating efficiency calculated based on unirradiated controls" (lines 88−91).We sincerely thank the reviewer for the critical comments.
- Results: The authors must draw up a table with the D10 and RBE values ​​for each cell line and discuss the results of the positive HPV cell line with an RBE greater than 3.
Response:
We thank the reviewer for the valuable comments. In accordance with the suggestion, the D10 and RBE values for each cell line were summarized as Table 2; discussion on the HPV-positive UM-SCC-104 cell line that showed an RBE greater than 3 was added as follows: "UM-SCC-104 showed an RBE greater than 3, which was exceptionally high as HPV-positive cell line. In fact, D10 for X-rays for this cell line was intermediate (i.e., 6.5 Gy), ranking fifth among the ten cell lines examined in this study. By contrast, D10 for carbon ions for this cell line was the lowest among the ten cell lines (i.e., 1.9 Gy). This indicates a specific sensitivity of UM-SCC-104 to carbon ions. Wang et al. reported that in the repair of DNA double-strand breaks (DSBs) induced by heavy ions, only the Ku-dependent non-homologous end joining (NHEJ), but not other DBS repair pathways, is impaired (DNA Repair 2008;7:725−733). Tang et al. reported upon establishment of the UM-SCC-104, that this cell line shows strong and diffuse expression for EGFR (Head and Neck 2012;34:1480−1491), a protein known to be involved in NHEJ by interacting with the catalytic subunit of DNA-dependent protein kinase (Das et al., Cancer Res 2007;67:5267–5274). These data together may suggest that UM-SCC-104 relies heavily on NHEJ in the repair of DSB (lines 200−209)."
- Discussion: the authors should put forward hypotheses concerning the difference in behaviour of positive and negative HPV cell lines in response to photon and carbon ion irradiation. Do the authors have the p53 status of each cell line? These data would be important for interpreting the results.
Response:
We sincerely thank the reviewer for the comment. According to the suggestion, the p53 status of the cell lines used in this study was summarized as Table 3. The hypothesis on the different behavior of HPV positive and negative cell lines in response to photons and carbon ions associated with p53 status was put as follows: "There was obvious difference in photon sensitivity between HPV-positive and HPV-negative HNSCC lines examined in this study, whereas the difference in carbon ion sensitivity between the two groups were mediocre. It is worth noting that the HPV-positive lines harbor wild-type p53, whereas the HPV-negative lines were p53-mutant (Table 3). We have previously shown that photon sensitivity is affected by p53 status; functional deficiency in p53 contributes to photon resistance through abrogation of radiation-induced apoptosis. By contrast, carbon ions exert cell killing regardless of p53 status through efficient induction of mitotic catastrophe (Amornwichet et al., PLOS ONE 2014;9:e115121). Thus, it can be hypothesized that CIRT shows promising anti-tumor effect for anti-apoptotic p53-mutated HPV-negative HNSCCs, that should be tested in future research (lines 210−218)."
Reviewer 2 Report
In this study, the authors performed clonogenic assays to evaluate the relative biological effectiveness of carbon ion beams on HPV-positive and HPV-negative cell lines. The methods are appropriate and the results are interesting to heavy ion radiotherapy. The article is suitable for publication.
Author Response
Reviewer #2
In this study, the authors performed clonogenic assays to evaluate the relative biological effectiveness of carbon ion beams on HPV-positive and HPV-negative cell lines. The methods are appropriate and the results are interesting to heavy ion radiotherapy. The article is suitable for publication.
Response:
We sincerely thank the reviewer for evaluating our manuscript and for the encouraging comments.
Reviewer 3 Report
Well-designed and conducted study with interesting findings that add to the body of knowledge about carbon-ion therapy. I have some queries regarding presentation of results which would further strengthen the quality of this work.
- Amend x-axes in Figure 1a and 1b (Gy) and y-axes in Figure 2a and 2b (D10 Gy) to the same maxima between xray and carbon results. Consistency would help with interpretation of findings across techniques
- Table S1, are differences in alpha/beta values across techniques statistically significant? A simple non-parametric paired test can answer this question and would enable readers to see whether statement at line 122 is true. Please perform this analysis and amend results to reflect outcome
- Lines 144-146, change wording 'diagnosed as cancer' to 'diagnosed with cancer'
Author Response
Reviewer #3
Well-designed and conducted study with interesting findings that add to the body of knowledge about carbon-ion therapy. I have some queries regarding presentation of results which would further strengthen the quality of this work.
Response:
We sincerely thank the reviewer for evaluating our manuscript and for the encouraging comments. We made a thorough revision on our manuscript in accordance with the suggestions as follows.
Amend x-axes in Figure 1a and 1b (Gy) and y-axes in Figure 2a and 2b (D10 Gy) to the same maxima between xray and carbon results. Consistency would help with interpretation of findings across techniques.
Response:
Figures 1a, 1b, 2a, and 2b were revised in accordance with the suggestion. We thank the reviewer for the precious comment that improved the quality of our manuscript.
Table S1: are differences in alpha/beta values across techniques statistically significant? A simple non-parametric paired test can answer this question and would enable readers to see whether statement at line 122 is true. Please perform this analysis and amend results to reflect outcome.
Response:
We thank the reviewer for the valuable comment. The differences in alpha/beta values across techniques were not statistically significant (P = 0.46) according to non-parametric two-sided Wilcoxon paired signed rank test. The corresponding text was revised as follows "the differences in a/b values between X-rays and carbon ions were not statistically significant (P = 0.46)" (lines 130−131). The statistical methods were added (lines 103−104).
Lines 144-146, change wording 'diagnosed as cancer' to 'diagnosed with cancer'.
Response:
We thank the reviewer for pointing out the grammatical error. The description was corrected accordingly (line 156).